# Prediction of Late-Onset Small for Gestational Age and Fetal Growth Restriction by Fetal Biometry at 35 Weeks and Impact of Ultrasound–Delivery Interval: Comparison of Six Fetal Growth Standards

**DOI:** 10.3390/jcm10132984

**Published:** 2021-07-03

**Authors:** Ricardo Savirón-Cornudella, Luis Mariano Esteban, Rocío Aznar-Gimeno, Peña Dieste-Pérez, Faustino R. Pérez-López, Jose Manuel Campillos, Berta Castán-Larraz, Gerardo Sanz, Mauricio Tajada-Duaso

**Affiliations:** 1Department of Obstetrics and Gynecology, Hospital Clínico San Carlos and Instituto de Investigación Sanitaria San Carlos (IdISSC), Calle del Prof Martín Lagos s/n, 28040 Madrid, Spain; rsaviron@gmail.com; 2Department of Applied Mathematics, Escuela Universitaria Politécnica de La Almunia, Universidad de Zaragoza, Calle Mayor 5, 50100 La Almunia de Doña Godina, Spain; 3Department of Big Data and Cognitive Systems, Instituto Tecnológico de Aragon, ITAINNOVA, María de Luna 7-8, 50018 Zaragoza, Spain; raznar@itainnova.es; 4Department of Obstetrics and Gynecology, Health Research Institute (IISA), Miguel Servet University Hospital and Aragon, Isabel La Católica 3, 50009 Zaragoza, Spain; pdpe88@gmail.com (P.D.-P.); jmcampillos@salud.aragon.es (J.M.C.); 5Department of Obstetrics and Gynecology, Faculty of Medicine and Instituto de Investigación Sanitaria Aragón, University of Zaragoza, Domingo Miral s/n, 50009 Zaragoza, Spain; faustino.perez@unizar.es; 6Department of Obstetrics and Gynecology, San Pedro Hospital, Calle Piqueras 98, 26006 Logroño, Spain; bcastan@riojasalud.es; 7Department of Statistical Methods and Institute for Biocomputation and Physics of Complex Systems-BIFI, University of Zaragoza, Calle Pedro Cerbuna 12, 50009 Zaragoza, Spain; gerardo.sanz@unizar.es; 8Department of Obstetrics and Gynecology, Miguel Servet University Hospital, Isabel La Católica 3, 50009 Zaragoza, Spain; mtajadad@gmail.com

**Keywords:** adverse perinatal outcomes, birth weight, estimated fetal weight, estimated percentile weight, fetal growth standard, small for gestational age, ultrasound

## Abstract

Small-for-gestational-age (SGA) infants have been associated with increased risk of adverse perinatal outcomes (APOs). In this work, we assess the predictive ability of the ultrasound-estimated percentile weight (EPW) at 35 weeks of gestational age to predict late-onset SGA and APOs, according to six growth standards, and whether the ultrasound–delivery interval influences the detection rate. To this purpose, we analyze a retrospective cohort study of 9585 singleton pregnancies. EPWs at 35 weeks were calculated to the customized Miguel Servet University Hospital (MSUH) and Figueras standards and the non-customized MSUH, Fetal Medicine Foundation (FMF), INTERGROWTH-21st, and WHO standards. As results of our analysis, for a 10% false positive rate, the detection rates for SGA ranged between 48.9% with the customized Figueras standard (AUC 0.82) and 60.8% with the non-customized FMF standard (AUC 0.87). Detection rates to predict SGA by ultrasound–delivery interval (1–6 weeks) show higher detection rates as intervals decrease. APOs detection rates ranged from 27.0% with FMF to 7.9% with the Figueras standard. In conclusion, the ability of EPW to predict SGA at 35 weeks is good for all standards, and slightly better for non-customized standards. The APO detection rate is significantly greater for non-customized standards.

## 1. Introduction

Screenings for fetal growth abnormalities are essential components of antenatal care, and fetal ultrasound plays a key role in the assessment of these conditions [1,2,3]. Small-for-gestational-age (SGA) infants—those with a birth weight below the 10th percentile according to the standards [4]—have been associated with increased risk of adverse perinatal outcomes (APOs) [5].

These fetuses are the leading cause of stillbirth [6,7,8], and have more risks of both neonatal morbidity [9] and mortality [10,11]. Recent studies have shown that an early diagnosis of SGA in the third trimester can help to reduce APOs, reflecting the benefit of prenatal diagnosis in these cases [12,13], although the time to perform the ultrasound is not clearly established.

Several studies customized or not to maternal and fetal physiological variables have been proposed to predict SGA [14]; these fetal growth standards [15,16,17] are based on Hadlock et al.’s methodology [18,19], or on new multilevel models [20,21,22], and can be customized to maternal and fetal physiological variables [23]. The estimation of the percentile adjusted for maternal and fetal characteristics is the property that postulates customized standards as better detectors of adverse perinatal outcomes than population-based standards (non-customized (NC)) [24]. The Royal College of Obstetricians and Gynecologists (RCOG) [25] recommends the use of customized birthweight curves to identify SGA fetuses; the adjustment of fetal weight should be performed individually, and not by population—although some studies have questioned the superiority of the EPW by customized standards and its association with APOs [26,27], and SGA with APOs [28].

Furthermore, new standards have recently been published by EFW [29], including international standards from the World Health Organization (WHO) [17] and the INTERGROWTH-21st project [20,21], and local standards from the Fetal Medicine Foundation (FMF) [22]. According to the recent review by McCowan et al. (2018), international population ultrasound standards still require more comparative studies for validation [5].

Since the controversy arose about the most appropriate method to predict SGA, and the lack of comparative assessment for the cited approaches, the objective of this study is to compare the ability of EPW—according to six growth standards, by ultrasound at 35 weeks, including population, population-customized, and international references—to predict late-onset SGA, defined as a birth weight below the 10th percentile at term delivery. The secondary objective is to determine whether the ultrasound–delivery interval influences the detection rate of SGA newborns.

## 2. Materials and Methods

### 2.1. Study Design

This was a retrospective cohort study of births assisted at the Miguel Servet University Hospital (MSUH), between March 2012 and December 2016. The inclusion criteria were as follows: live singleton pregnancies controlled in MSUH from the first trimester of gestation; fetal ultrasound assessment at gestational age of 35 (range 34–36) weeks; and deliveries between 37 and 42 weeks of gestational age of fetuses without stillbirth associated with malformations or chromosomal abnormalities. Of the 19,310 consecutive deliveries assisted in our hospital in the period studied, the 9585 cases that fulfilled the specific inclusion criteria—such as data availability to estimate percentile weights by standards—were considered for the analysis. Study participants’ selection samples are detailed in Figure 1.

The last menstrual period was adjusted by first trimester ultrasound [30]. Universal ultrasound screening was performed at 35 weeks (range 34–36 weeks) at the Ultrasound and Prenatal Diagnosis Unit using either a Voluson 730 Expert, E6, E8 ultrasound machine (General Electric, Healthcare, Zipf, Austria) or an Aloka Prosound SSD-5000 (Hitachi Aloka Medical Systems, Tokyo, Japan). This ultrasound corresponds to the one that is routinely performed in all pregnancies at our center to try to increase the detection of fetal growth alterations [5].

EFW was calculated with the formula of each standard to which it was built. We used Hadlock et al.’s [19] formula, which combines biparietal diameter, cephalic and abdominal circumference, and femur length, for the MSHU, Figueras et al., and WHO standards; and the version that uses cephalic and abdominal circumference and femur length for the Fetal Medicine Foundation (FMF) standard. In addition, Stirnemann et al.’s formula [20], including only cephalic and abdominal circumference, was used to estimate percentile weight for the INTERGROWTH-21st standard.

For the calculation of the EPW, we collected in the study the maternal age and body mass index (BMI) at the beginning of pregnancy, parity, maternal and paternal height, maternal ethnic origin, smoking habits, infant gender, birth weight, and ultrasound EFW. We also collected perinatal outcomes in order to analyze APOs in SGA infants at delivery, defined as the occurrence of a 5-min Apgar score < 7, instrumental or cesarean delivery for non-reassuring fetal status, arterial cord blood pH < 7.10, and stillbirth.

### 2.2. Estimated Percentile Weight

EPWs were calculated according to 6 different customized and NC growth standards, including population, population-customized, and international references. For the customized standards, the methodologies of Hadlock et al. [18] and Gardosi et al. [23] were used for (1) the MSUH standard customized for parity, age, BMI, maternal height, paternal height, and fetal gender, built using a modified version of Hadlock et al.’s growth charts adjusted to our population, with a coefficient of variation that changes with gestational age (Saviron-Cornudella et al. [16]); (2) and the Barcelona Clinic Hospital (Figueras et al. [15]). For the NC standards, we used (3) an NC version of the MSUH standard (Saviron-Cornudella et al. [16]); (4) the international population INTERGROWTH–21st [20,21]—a multilevel mixed model whose main characteristic is that it includes pregnant women without pathology; (5) the international WHO fetal growth standard [17], and (6) the FMF local growth multilevel mixed model (Nicolaides at al [22]).

To assess ultrasound weight measures in the third trimester, EPWs were estimated between 34 and 36 weeks of gestational age. The WHO EPW was calculated by interpolation of the 5th, 10th, 25th, 50th, 75th, 90th, and 95th percentiles.

As a gold standard for the analysis, SGA was defined as a birth weight below the 10th percentile, using a growth reference for the Spanish population based on 9362 birthweights [31]. We did not focus our analysis on intrauterine growth-restricted fetuses (IUGRs). As we did not perform Doppler ultrasound universally (only in cases of estimated fetal weight < 10th percentile), we did not study the subgroup of SGA fetuses at delivery with altered Doppler ultrasound. This is because a significant percentage of SGA fetuses at delivery did not present an estimated fetal weight <10th percentile by ultrasound.

### 2.3. Statistical Analysis

Data were descriptively analyzed using medians and interquartile ranges for continuous variables, and absolute and relative frequencies for categorical variables. The ability of EPW provided by the six standards to predict SGA was analyzed using the area under the receiver operating characteristic curve (AUC) [32]. Sensitivity (detection rate) was established for false positive rates (FPR) of 5, 10, 15, and 20%. The percentile threshold point corresponding to the FPR values was also calculated. AUCs were compared using the DeLong test, and sensitivities through a proportion comparison test.

In addition, we built logistic regression models to estimate the OR and 95% confidence interval that correspond to an increase of 1% in the EPW at 35 weeks, as a predictor for SGA at delivery, performing a subanalysis for different ultrasound–delivery intervals (1–6 weeks).

We analyzed the diagnostic ability of the EPW 10th percentile and SGA birthweights to detect the following adverse perinatal outcomes: 5-min Apgar score < 7, instrumental delivery for non-reassuring fetal status (NRFS), cesarean delivery for NRFS, arterial cord blood pH < 7.10, and stillbirth. Comparison between APOs predicted by standards was performed using a proportion test.

Analyses were performed using R version 3.6.2 language programming (The R Foundation for Statistical Computing, Vienna, Austria) [33].

## 3. Results

### 3.1. Descriptive Results

Table 1 shows the descriptive characteristics of the pregnant women, and also displays medians and percentiles 10 (P10) and 90 (P90) among groups for the six studied standards for EFWs by ultrasound at 35 weeks (range from 34+0 to 36+6 weeks). WHO and FMF standards show an underestimation of the median expected value (50%) by ultrasound (median values 43.1%, P10–P90 range 7.5–74.9, and 37.6%, P10–P90 range 2.7–89.9, respectively), while the Figueras standard shows an overestimation by ultrasound (median values 59.3%, P10–P90 range 18.1–93.5).

EPW distributions are detailed in Figure 2, where a comparison of the percentage of SGA is shown for each standard. The rate of SGA at birth in our cohort was 9.4% (*n* = 902).

Regarding APOs, Table 2 shows that SGA deliveries included 21.6% (*n* = 139) APOs, 28.6% (*n* = 12) 5-min Apgar scores < 7, 19.9% (*n* = 32) instrumental deliveries for NRFS, 26.8% (*n* = 71) cesarean deliveries for NRFS, 17.7% (*n* = 45) neonatal acidemia (pH cord blood pH < 7.10), and 26.3% (*n* = 5) stillbirth.

### 3.2. Comparison of Standards

Table 3 displays values of AUCs and sensitivities plus the percentile threshold points for different FPRs to predict SGA at delivery by ultrasound at 35 weeks (range 34+0–36+6 weeks). For a 10% FPR, the detection rates for SGA for all standards ranged between 48.9% with the Figueras standard (AUC: 0.82; 95% CI: 0.80–0.83) to 60.8% with the Fetal Medicine Foundation standard (AUC: 0.87; 95% CI: 0.85–0.88). These values were obtained with percentile threshold points below 17.3%, 16.3%, 22.9%, 17.3%, 11.1% and 5.3% for NC MSUH, customized MSUH, Figueras, INTERGROWTH-21st, WHO, and FMF standards, respectively. For a 20% FPR, the detection rates were between 66.4 and 78.92%, using 28.5%, 28.1%, 34.6%, 28.1%, 19.1%, and 13.1% as percentile threshold points for the abovementioned standards, respectively.

Figure 3 illustrates the receiver operating characteristic curve comparison and AUC for the prediction of SGA at delivery by ultrasound at 35 weeks. Figure 4 displays the results of the logistic regression model with the ORs and 95% CIs to predict SGA by ultrasound at 35 weeks, according to the standards.

*p*-values of the comparisons of the standard AUC values and sensitivity for a 90% specificity are shown in Table 4. The Fetal Medicine Foundation and the non-customized MSUH standards showed no statistically significant differences between them, with greater SGA prediction ability than the Intergrowth-21st, Figueras, and WHO standards. Moreover, in the comparison, the Intergrowth-21st and WHO standards showed significant differences from the customized MSHU and Figueras standards. Finally, customized standards did not show differences between them.

Regarding APO prediction by EPW < 10, in Table 2 we show that the Fetal Medicine Foundation and WHO standards reached the greatest detection rates—27.0% and 17.4% respectively—with statistically significant differences between them and the rest of standards. No statistically significant differences were detected in the any of the possible comparisons between the non-customized MSUH (11.8%), INTERGROWTH-21st (10.7%), customized MSUH (9.6%), and Figueras (7.9%) standards, with the unique exception of the significant difference between the non-customized MSUH and Figueras standards. The instrumental deliveries for NRFS, cesarean deliveries for NRFS, and neonatal acidemia APOs might explain those differences. *p*-values of all comparisons are illustrated in Table 4.

### 3.3. Ultrasound-Delivery Interval: Comparison of Standards

Table 5 displays values of AUCs and sensitivities for different FPRs to predict SGA by ultrasound–delivery interval (range 1–6 weeks). The observed results show higher detection rates as the interval decreases. Figure 5 shows the prediction of small for gestational age cases, by standard, by ultrasound–delivery interval (1–6 weeks), for a 10% false positive rate.

Figure 6 displays odds ratios and 95% confidence intervals of the standards in order to predict SGAs by ultrasound–delivery interval (range 1–6 weeks). Figure 7 illustrates the receiver operating characteristic curve comparison of fetal growth standards for the prediction of SGA newborns according to the ultrasound–delivery interval (range 1–6 weeks).

## 4. Discussion

### 4.1. Principal Findings

We have demonstrated the utility of EPW by ultrasound at 35 weeks (range 34+0–36+6 weeks) as a predictor of SGA fetuses at delivery at term. Adjusting the percentile threshold points, the growth standards showed a similar good predictive ability, but with a significant advantage for the non-customized MSUH and Fetal Medicine Foundation standards, and a disadvantage for both the customized MSUH and Figueras standards, for SGA fetuses.

In our results, we found an underestimation of 10th percentile, by ultrasound at 35 weeks (range 34–36 weeks) with the WHO (7.5%) and FMF (2.7%) standards, and an overestimation with the Figueras (18.1%) standard; for that, we can conclude that these standards have a lack of calibration for our study population. The MSHU (NC (11.9%) and customized (12.2%)) and INTERGROWTH-21st (12.7%) standards fit better to the 10th percentile, with a minimum error, probably for the exclusion of premature deliveries.

When we analyzed the APO-predictive ability of the six standards by percentile weight <10 at 35th week of gestational age, the customized Fetal Medicine Foundation and WHO standards showed the greatest diagnostic ability, with statistically significant differences from the rest of standards. The main reason for this lies in the greater proportion of 10th percentile EPW for the Fetal Medicine Foundation (21.2%) and WHO (12.6%) standards, In any case, with similar proportions of EPW < 10, the non-customized MSUH and INTERGROWTH-21st standards show a better APO-predictive ability than the customized MSUH and Figueras standards. A previous study did not find any significant differences between the customized and non-customized standards when analyzing the predictive ability of EPW to detect APOs; by contrast, using EPW > 90th percentile, we detected significant differences [34].

### 4.2. Prediction by Fetal Biometry and Ultrasound–Delivery Interval

There is no international consensus on performing a universal ultrasound in the third trimester; two international guidelines—the RCOG [35], and the American College of Obstetrics and Gynecology (ACOG) [36]—do not recommend universal ultrasound to detect fetal growth anomalies. Sovio el al [37], however, found that universal third trimester ultrasound in nulliparous women, compared with selected ultrasound, tripled the detection of SGA < P10 infants, and could identify FGR fetuses at increased risk of neonatal morbidity.

The EPW at third trimester ultrasound over 32 weeks has been shown to be a good predictive model (AUC > 0.85) for the detection of SGA at delivery in several studies, although with detection rates limited for late-onset SGA births [4,38,39]. For gestational time, the detection rate of SGA at delivery by ultrasound between 33–34 weeks is approximately 52%, and between 36–37 weeks it is approximately 60% (FPR of 10%) [40,41,42]. According to several studies, therefore, detection is higher the later the ultrasound is performed [14,43,44]. In our case, the predictive capacity for SGA at delivery by ultrasound at 35 weeks is also limited for the six growth standards, and generally, a shorter ultrasound–delivery interval is correlated with better prediction rates. In any case, the cutoff points of the 10th percentiles by ultrasound at 35 weeks are moderate for the prediction of SGA at delivery.

### 4.3. Prediction by Fetal Biometry and Ultrasound–Delivery Interval: Comparison of Standards

Blue et al., in 2018 [45], compared the RCOG and ACOG standards for the detection of SGA at delivery, with a mean birth of 37.7 weeks and ultrasounds performed in the previous 2 weeks, and showed that both standards had a moderate predictive capacity (AUCs of 0.78 and 0.76, respectively). In another study by Blue in 2019 [46], the Hadlock and INTERGROWTH-21st standards for the detection of SGA, with deliveries at 37 weeks on average and ultrasound in the previous two weeks, showed good predictive capabilities (>0.90), with cutoff points of the optimal percentile at 15% for the Hadlock standard and 22% for the INTERGROWTH-21st standard. Both studies are not comparable to ours; although they show the minimum differences in SGA prediction regardless of the standard used, neither of them studied customized standards.

In two studies by Odibo et al. in 2018 [47] and 2019 [48], using the same sample obtained for the three different standards compared (INTERGROWTH-21st, a local customized standard, and the Hadlock standard), a moderate predictive capacity for SGA at delivery was achieved (0.67, 0.62, and 0.69, respectively), although with ultrasound performed between 26 and 36+6 weeks, and an average ultrasound–delivery interval of 6.7 weeks—also different from our study.

Reboul et al., in 2017 [49], found that the Hadlock and the customized Gardosi standards had a moderate predictive capacity for SGA at delivery (0.768 and 0.708, respectively), with the detection rates somewhat higher for the Hadlock standard, although with an average of performing ultrasound at 32 weeks—lower than ours, which could justify the lower predictive capacity.

### 4.4. Clinical and Research Implications

In clinical practice we can say that more important than the choice of the growth standard is its calibration before clinical use, both by ultrasound and delivery, in the reference population. The physiological and non-pathological characteristics of each population are those that will allow us to calibrate the standard to be used.

There are several factors for which ultrasound in the third trimester presents limitations when predicting SGA and FGRs at delivery, and some of them are unavoidable—especially the systematic error of ultrasound at the time of EFW calculation [50]. With the current studies carried out on the timing of performing the third trimester ultrasound and the ultrasound–delivery interval, together with our comparative study of standards, we can affirm that the timing that better predicts SGA cases is the one closest to delivery; however, we cannot delay ultrasound universally to 37 weeks, since we would not detect early FGRs. As we are not currently able to make that prediction, it will continue to be the subject of future research.

According to our results, it would be appropriate to raise the ultrasound-estimated weight percentile cutoff point above 10 for fetal growth control. This is because the 10th percentile has been shown to be insufficient, and with low predictive capacity for SGA at delivery and, therefore, fetuses that can potentially be IUGR even before delivery can escape control and, thus, increase their morbidity and mortality. Our recommendation, in the ultrasound during the third trimester, between 35 and 36 weeks, could be to raise the cutoff point at least from the 10th to the 20th percentile for strict control of fetal growth.

### 4.5. Strengths and Limitations of the Study

Our study has several strengths, including the wide sample size close to 10,000 pregnancies. Ultrasound measurements were performed in routine clinical practice; thus, weight estimations were more concentrated over specific weeks of gestational age. Limitations of our investigation are that our data came from a single hospital, and their retrospective nature could limit the generalization of our standards. Furthermore, the information of the ultrasound was available to the obstetricians, which could mean a bias in the management of the pregnancies. A small percentage of labors are inductions of labor or cesarean sections programmed by IUGR, and they could act as confounding factors in the study. Similarly, other cases of early termination due to other causes have not been taken into account.

## 5. Conclusions

In summary, even with limited detection rates, the growth standards showed a similar good predictive ability, with a statistically significant improvement by the use of non-customized standards, for SGA at delivery by ultrasound at 35 weeks. Generally, a shorter ultrasound delivery interval for the different standards was correlated with better prediction rates for small gestational age cases. When focusing on the use of EPW < 10th percentile at week 35 for the prediction of APOs, non-customized standards also demonstrated an advantage over customized standards.

## Figures and Tables

**Figure 1 jcm-10-02984-f001:**
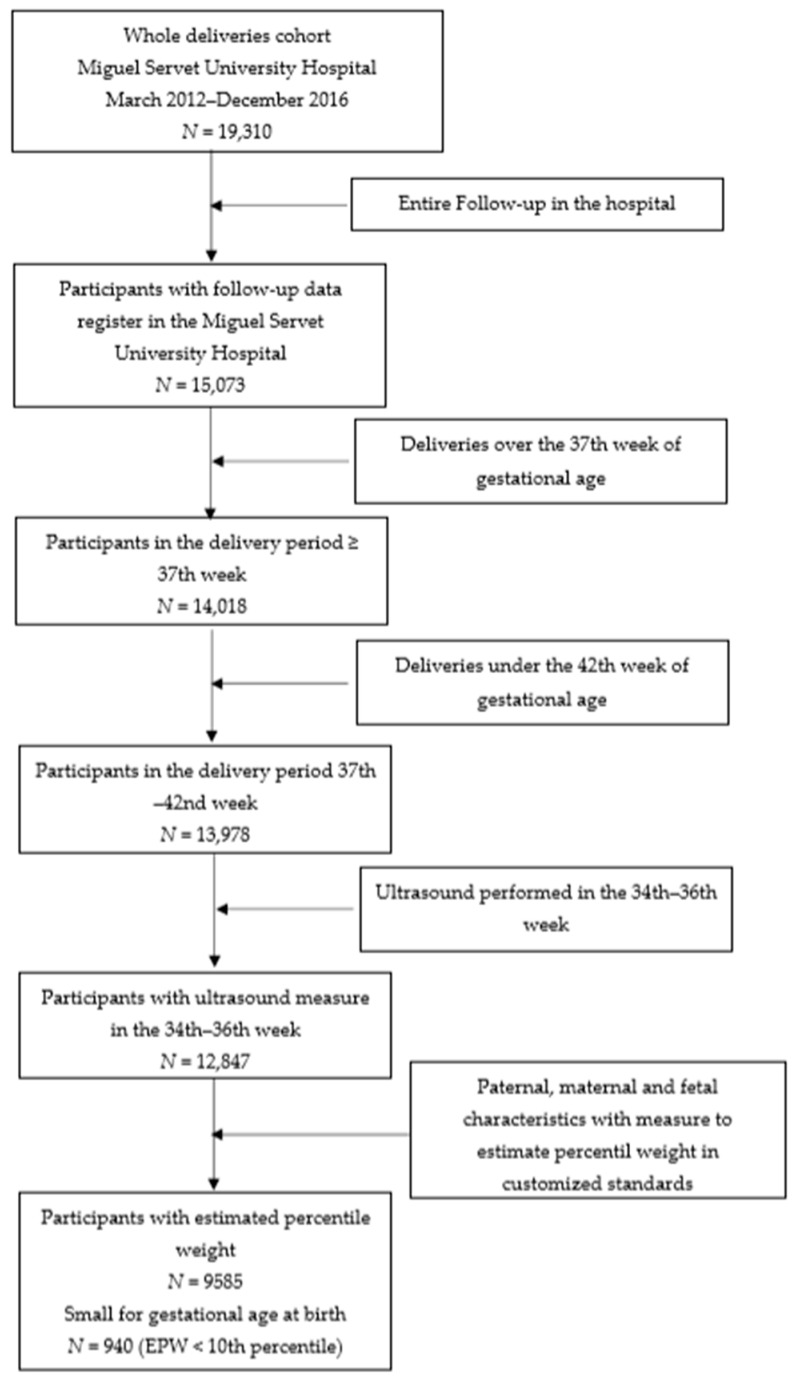
Flowchart of patient recruitment; EPW: estimated percentile weight.

**Figure 2 jcm-10-02984-f002:**
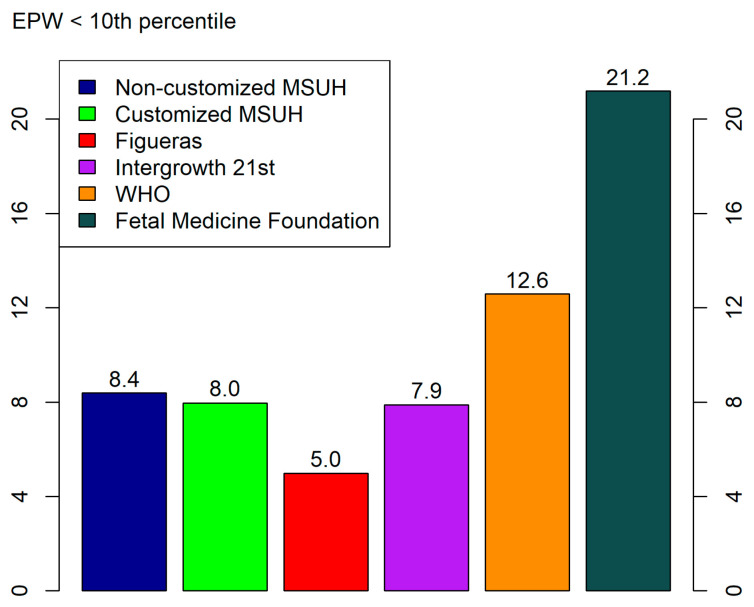
Percentage of small for gestational age (estimated percentile weight (EPW) <10th percentile) cases provided by standards at the third trimester (34th–36th week). Growth standards: non-customized Miguel Servet University Hospital (MSUH)16, customized MSUH, Figueras et al., INTERGROWTH-21st, World Health Organization (WHO), and Fetal Medicine Foundation.

**Figure 3 jcm-10-02984-f003:**
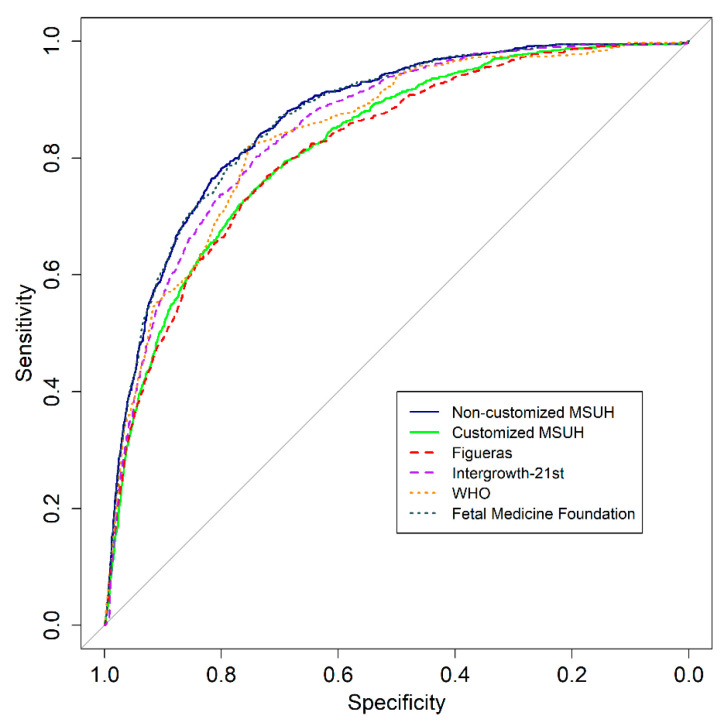
Receiver operating characteristic curves: comparison of fetal growth standards and area under the curve (95% CIs) for prediction of small for gestational age newborns, using estimated percentile weight by ultrasound at 35 weeks. Growth standards: non-customized Miguel Servet University Hospital (MSUH), customized MSUH, Figueras et al., INTERGROWTH-21st, World Health Organization (WHO), and Fetal Medicine Foundation.

**Figure 4 jcm-10-02984-f004:**
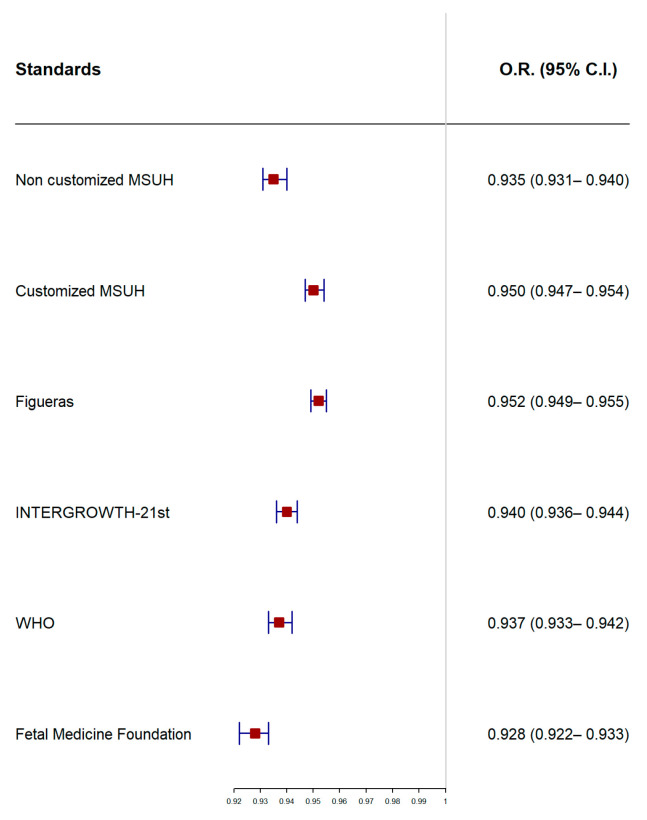
Odds ratios (ORs) and 95% confidence intervals (CIs) of standards in to predict small for gestational age according to estimated percentile weight by ultrasound at 35 weeks; MSUH: Miguel Servet University Hospital; WHO: World Health Organization.

**Figure 5 jcm-10-02984-f005:**
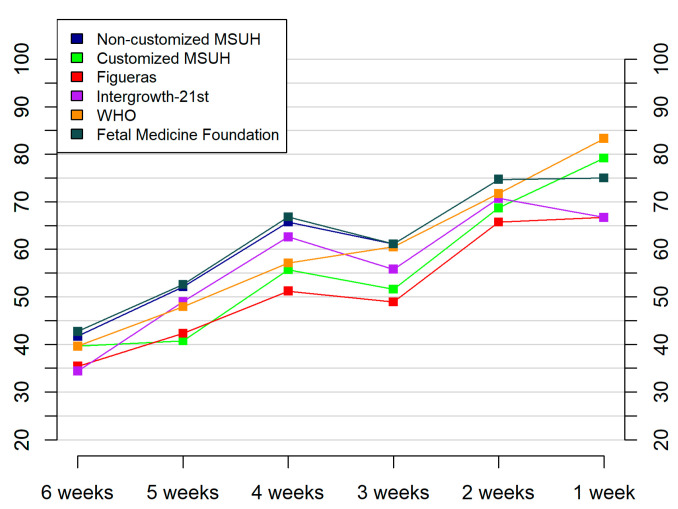
Prediction of small for gestational age cases, by standard and by ultrasound–delivery interval (1–6 weeks), for a 10% false positive rate. Growth standards: non-customized Miguel Servet University Hospital (MSUH), customized MSUH, Figueras et al., INTERGROWTH-21st, World Health Organization (WHO), and Fetal Medicine Foundation.

**Figure 6 jcm-10-02984-f006:**
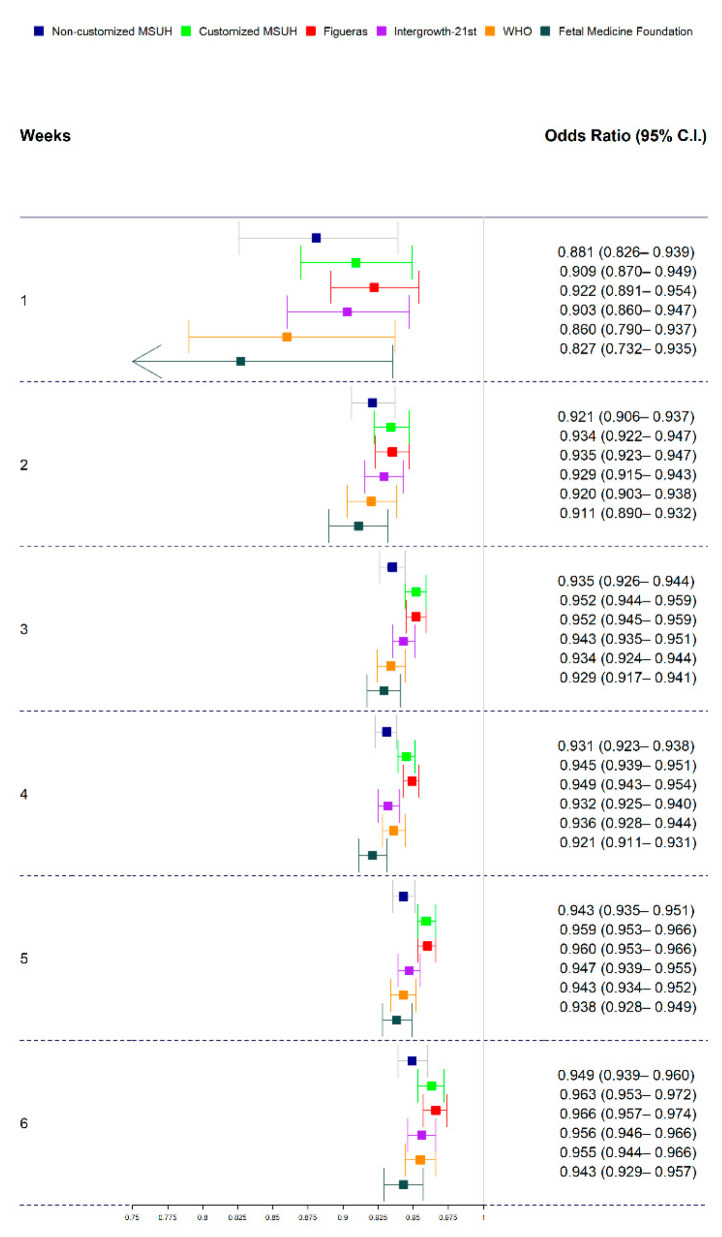
Odds ratios (ORs), 95% confidence intervals (CIs), and *p-*values of standards, in order to predict small for gestational age (SGA) fetuses by standard and ultrasound–delivery interval delivery date (1–6 weeks). Growth standards: non-customized Miguel Servet University Hospital (MSUH), customized MSUH, Figueras et al., INTERGROWTH-21st, World Health Organization (WHO), and Fetal Medicine Foundation.

**Figure 7 jcm-10-02984-f007:**
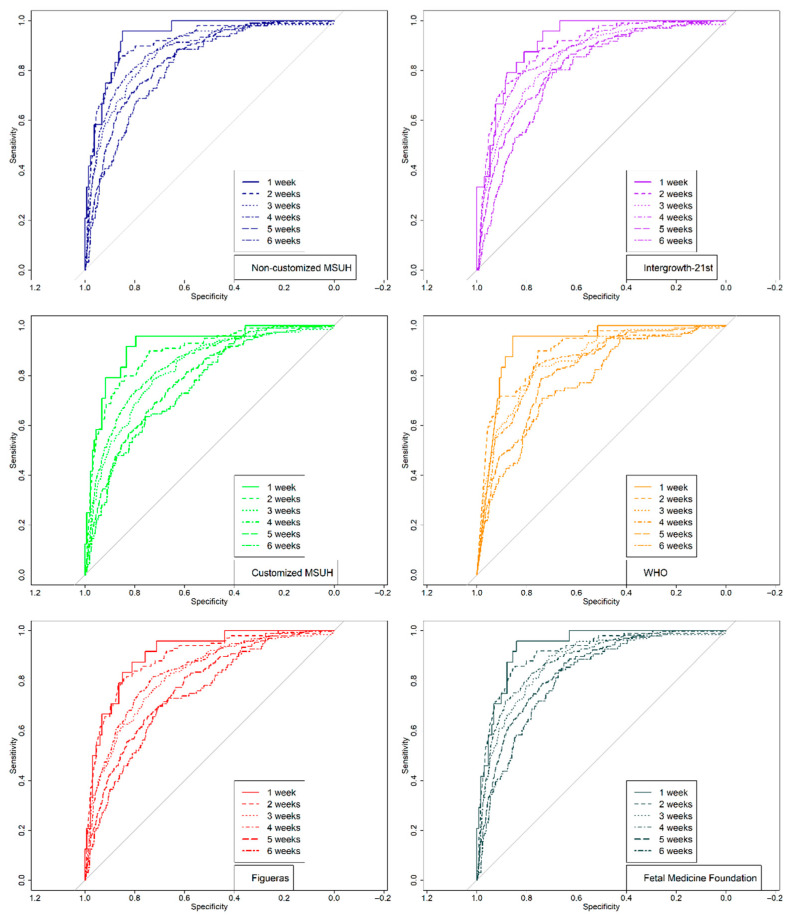
Receiver operating characteristic curves: comparison of fetal growth standards and area under the curve for prediction of small for gestational age (SGA) by standard and by ultrasound–delivery interval (1–6 weeks). Growth standards: non-customized Miguel Servet University Hospital (MSUH), customized MSUH, Figueras et al., INTERGROWTH-21st, World Health Organization (WHO), and Fetal Medicine Foundation.

**Table 1 jcm-10-02984-t001:** Parental baseline characteristics (top), pregnancy (middle), and perinatal characteristics (bottom) of pregnancies. Data are reported as n (%) or medians (interquartile range); MSUH: Miguel Servet University Hospital; NRFS: non-reassuring fetal status; WHO: World Health Organization.

Clinical Characteristics	Pregnancies (*n* = 9585)
*Parental characteristics*	
Maternal age (years)	33.3 (30.1–36.1)
Maternal body mass index (kg/m^2^)	23.2 (21.1–26.2)
Maternal height (cm)	163 (159–168)
Paternal height (cm)	176 (172–181)
Parity	
0	5077 (53.0%)
1	3724 (38.9%)
≥ 2	784 (8.1%)
Maternal ethnicity	
Caucasian	9243 (96.4%)
Asian	110 (1.1%)
African	232 (2.4%)
Maternal smoking habits	
Yes	1546 (16.1%)
No	8039 (83.9%)
*Ultrasound parameters at 35 (34–36) weeks*	
Gestational age (weeks) at ultrasound	35.1 (35.0–35.3)
Estimated fetal weight (grams) by Hadlock	2495 (2314–2697)
Estimated fetal weight (grams) by Stirnemann	2421 (2209–2648)
Percentile by standard	P50 (P10–P90)
Non-customized MSUH	52.6 (11.9–93.3)
Customized MSUH	52.9 (12.2–92.9)
Figueras	59.3 (18.1–93.5)
INTERGROWTH-21st	51.9 (12.7–89.8)
WHO	43.1 (7.5–74.9)
Fetal Medicine Foundation	37.6 (2.7–89.9)
*Pregnancy and perinatal outcomes*	
Gestational age at delivery	40.0 (39.1–40.7)
Newborn gender	
Female	4652 (48.5%)
Male	4933 (51.5%)
Birth weight	3310 (3030–3590)
Small for gestational age (<10th percentile)	902 (9.4%)
5-min Apgar score < 7	42 (0.4%)
Instrumental delivery for NRFS	161 (1.7%)
Cesarean delivery for NRFS	265 (2.8%)
Arterial cord blood pH < 7.10	254 (2.6%)
Stillbirth	19 (0.2%)
Any adverse perinatal outcome *	645 (6.7%)

* Excluding SGA.

**Table 2 jcm-10-02984-t002:** Diagnosis of adverse perinatal outcomes (APOs); EPW: estimated percentile weight at 35 weeks (range 34–36 weeks); MSUH: Miguel Servet University Hospital; NRFS: non-reassuring fetal status; WHO: World Health Organization.

	5-Min Apgar Score < 7	Instrumental Delivery for NRFS	Cesarean Delivery for NRFS	Arterial Cord Blood pH < 7.10	Stillbirth	Any APO
Total cohort	42	161	265	254	19	645
SGA	12 (28.6%)	32 (19.9%)	71 (26.8%)	45 (17.7%)	5 (26.3%)	139 (21.6%)
EPW < 10						
Non-customized MSUH	8 (19.0%)	15 (9.3%)	43 (16.2%)	26 (10.2%)	5 (26.3%)	76 (11.8%)
Customized MSUH	6 (14.3%)	9 (5.6%)	38 (14.3%)	22 (8.7%)	5 (26.3%)	62 (9.6%)
Figueras	5 (11.9%)	8 (5.0%)	32 (12.1%)	18 (7.1%)	4 (21.1%)	51 (7.9%)
INTERGROWTH-21st	7 (16.7%)	11 (6.8%)	39 (14.7%)	26 (10.2%)	5 (26.3%)	69 (10.7%)
WHO	12 (28.6%)	24 (14.9%)	57 (21.5%)	42 (16.5%)	7 (36.8%)	112 (17.4%)
FMF	17 (40.5%)	37 (23.0%)	89 (33.6%)	62 (24.4%)	10 (52.6%)	174 (27.0%)

**Table 3 jcm-10-02984-t003:** Area under the receiver operating characteristic curve and sensitivity analyses to detect small for gestational age cases by ultrasound at 35 weeks (range 34–36 weeks) for different false positive rate (FPR) percentages; MSUH: Miguel Servet University Hospital; Pc: percentile; WHO: World Health Organization. * Sensitive threshold percentile (Thr): percentile point that corresponds to a false positive rate value.

Prediction of Small for Gestational Age by Standard	Area under the Curve (95% C.I.)	Sensitivity (95% C.I.) and Threshold Percentile Points *			
		FPR 5%	FPR 10%	FPR 15%	FPR 20%
**Small for gestational age**					
Non-customized MSUH	0.87 (0.85–0.88)	42.6 (39.4–45.9) (Thr: 10.3)	60.4 (57.1–63.6) (Thr: 17.3)	70.5 (67.4–73.4) (Thr: 23.3)	78.2 (75.3–80.8) (Thr: 28.5)
Customized MSUH	0.82 (0.80–0.83)	35.5 (32.3–38.6) (Thr: 9.9)	51.1 (47.8–54.4) (Thr: 16.3)	60.9 (57.6–64.1) (Thr: 22.7)	67.6 (64.4–70.6) (Thr: 28.1)
Figueras	0.82 (0.80–0.83)	35.4 (32.3–38.6) (Thr: 14.5)	48.9 (45.6–52.2) (Thr: 22.9)	60.8 (57.5–64.0) (Thr: 29.5)	66.4 (63.2–69.5) (Thr: 34.6)
INTERGROWTH-21st	0.85 (0.84–0.86)	37.8 (34.6–41.1) (Thr: 10.2)	56.3 (53.0–59.6) (Thr: 17.3)	66.7 (63.5–69.8) (Thr: 22.9)	73.7 (70.7–76.5) (Thr: 28.3)
WHO	0.84 (0.83–0.85)	38.6 (35.4–41.9) (Thr: 6.2)	56.1 (52.8–59.4) (Thr: 11.1)	61.0 (57.7–64.2) (Thr: 14.6)	70.6 (67.5–73.5) (Thr: 19.1)
Fetal Medicine Foundation	0.87 (0.85–0.88)	42.4 (39.2–45.7) (Thr: 1.9)	60.8 (57.5–64.0) (Thr: 5.3)	70.7 (67.6–73.6) (Thr: 9.3)	76.3 (73.4–79.0) (Thr: 13.1)

**Table 4 jcm-10-02984-t004:** Results of *p*-value tests to compare standards: area under the receiver operating characteristic curve (AUC) and sensitivities (specificity 90%) to predict small for gestational age; and percentage of adverse perinatal outcome (APO) diagnosis; NC: non-customized; C: customized; MSUH: Miguel Servet University Hospital; WHO: World Health Organization.

	Customized MSUH			Figueras			INTERGROWTH-21st			WHO			Fetal Medicine Foundation		
	AUC	Sens	APOs	AUC	Sens	APOs	AUC	Sens	APOs	AUC	Sens	APOs	AUC	Sens	APOs
NC MSUH	<0.001	<0.001	0.242	<0.001	<0.001	0.025	<0.001	0.086	0.597	<0.001	0.071	0.006	0.169	0.900	<0.001
C MSUH				0.053	0.375	0.325	<0.001	0.030	0.580	<0.001	0.037	<0.001	<0.001	<0.001	<0.001
Figueras							<0.001	0.002	0.103	<0.001	0.003	<0.001	<0.001	<0.001	<0.001
IG-21st										0.094	0.970	<0.001	<0.001	0.058	<0.001
WHO													<0.001	0.048	<0.001

**Table 5 jcm-10-02984-t005:** Area under the receiver operating characteristic curve and sensitivity analyses to predict small for gestational age newborns using estimated percentile weight by ultrasound at 35 weeks (range 34–36 weeks), for different false positive rate (FPR) percentages and ultrasound–delivery intervals (1–6 weeks); MSUH: Miguel Servet University Hospital; WHO: World Health Organization.

Prediction of Small for Gestational by Standard and Ultrasound–Delivery Interval	N	Area under the Curve (95% C.I.)	Sensitivity			
			FPR 5%	FPR 10%	FPR 15%	FPR 20%
**Non-customized MSUH**						
1 week (8–14 days)	156	0.94 (0.90–0.98)	58.3 (36.9–77.2)	75.0 (52.9–89.4)	92.0 (71.9–98.7)	96.0 (77.1–99.8)
2 weeks (15–21 days)	767	0.91 (0.88–0.94)	63.6 (53.3–72.9)	74.7 (64.8–82.7)	85.9 (77.1–91.8)	88.9 (80.6–94.1)
3 weeks (22–28 days)	1725	0.87 (0.84–0.90)	46.3 (39.1–53.7)	61.1 (53.7–68.0)	68.9 (61.7–75.3)	77.4 (70.7–83.0)
4 weeks (29–35 days)	2965	0.88 (0.86–0.90)	48.4 (42.5–54.3)	65.7 (59.9–71.1)	75.4 (69.9–80.2)	81.0 (75.9–85.3)
5 weeks (36–42 days)	2596	0.84 (0.81–0.87)	32.5 (26.1–39.6)	52.1 (44.8–59.3)	65.5 (58.3–72.1)	71.6 (64.6–77.7)
6 weeks (43–49 days)	1276	0.81 (0.77–0.85)	26.0 (17.8–36.1)	41.7 (31.9–52.2)	53.1 (42.7–63.3)	65.6 (55.1–74.8)
**Customized MSUH**						
1 week (8–14 days)	156	0.92 (0.86–0.98)	58.3 (36.9–77.2)	79.2 (57.3–92.1)	83.3 (61.8–94.5)	91.7 (71.6–98.6)
2 weeks (15–21 days)	767	0.89 (0.85–0.92)	56.6 (46.3–66.4)	68.7 (58.5–77.4)	77.8 (68.1–85.3)	79.8 (70.3–86.9)
3 weeks (22–28 days)	1725	0.82 (0.79–0.85)	35.3 (28.6–42.6)	51.6 (44.3–58.9)	58.4 (51.0–65.4)	68.4 (61.2–74.8)
4 weeks (29–35 days)	2965	0.84 (0.82–0.86)	40.1 (34.4–46.0)	55.7 (49.8–61.5)	66.8 (61.0–72.1)	73.0 (67.4–78.0)
5 weeks (36–42 days)	2596	0.78 (0.74–0.81)	25.3 (19.5–32.1)	40.7 (33.8–48.0)	52.6 (45.3–59.8)	58.8 (51.5–65.7)
6 weeks (43–49 days)	1276	0.76 (0.71–0.80)	21.9 (14.4–31.7)	39.6 (29.9–50.1)	47.9 (37.7–58.3)	55.2 (44.7–65.2)
**Figueras**						
1 week (8–14 days)	156	0.91 (0.85–0.96)	54.2 (33.3–73.9)	66.7 (44.7–83.6)	79.2 (57.3–92.1)	87.5 (66.5–96.7)
2 weeks (15–21 days)	767	0.89 (0.85–0.92)	56.6 (46.3–66.4)	65.7 (55.4–74.8)	79.8 (70.3–86.9)	83.8 (74.7–90.2)
3 weeks (22–28 days)	1725	0.82 (0.79–0.85)	37.4 (30.6–44.7)	48.9 (41.6–56.2)	61.6 (54.3–68.5)	65.8 (58.5–72.4)
4 weeks (29–35 days)	2965	0.83 (0.81–0.86)	37.0 (31.5–42.9)	51.2 (45.3–57.1)	63.0 (57.1–68.5)	72.3 (66.7–77.3)
5 weeks (36–42 days)	2596	0.77 (0.74–0.81)	26.8 (20.8–33.7)	42.3 (35.3–49.6)	50.5 (43.3–57.7)	57.7 (50.6–64.7)
6 weeks (43–49 days)	1276	0.74 (0.70–0.79)	20.8 (13.5–30.5)	35.4 (26.1–45.9)	42.7 (32.8–53.2)	51.0 (40.7–61.3)
**INTERGROWTH–21st**						
1 week (8–14 days)	156	0.92 (0.87–0.96)	37.5 (19.6–59.2)	66.7 (44.7–83.6)	79.2 (57.3–92.1)	87.5 (66.5–96.7)
2 weeks (15–21 days)	767	0.89 (0.86–0.92)	54.5 (44.2–64.5)	70.7 (60.6–79.2)	76.8 (67.0–84.4)	81.8 (72.5–88.6)
3 weeks (22–28 days)	1725	0.84 (0.81–0.87)	38.4 (31.5–45.7)	55.8 (48.4–62.9)	63.7 (56.4–70.5)	72.6 (65.6–78.7)
4 weeks (29–35 days)	2965	0.87 (0.85–0–89)	41.5 (35.8–47.4)	62.6 (56.7–68.1)	73.3 (67.7–78.2)	79.9 (74.7–84.3)
5 weeks (36–42 days)	2596	0.82 (0.79–0.85)	32.5 (26.1–39.6)	49.0 (41.8–56.2)	61.3 (54.0–68.1)	68.0 (60.9–74.4)
6 weeks (43–49 days)	1276	0.79 (0.74–0.83)	17.7 (10.9–27.1)	34.4 (25.2–44.9)	51.0 (40.7–61.3)	56.2 (45.7–66.2)
**WHO**						
1 week (8–14 days)	156	0.92 (0.87–0.97)	42.5 (23.5–63.8)	83.3 (61.8–94.5)	95.8 (76.8–99.8)	95.8 (76.8–99.8)
2 weeks (15–21 days)	767	0.89 (0.86–0.93)	59.6 (49.2–69.2)	71.7 (61.6–80.1)	71.7 (61.6–80.1)	78.8 (69.2–86.1)
3 weeks (22–28 days)	1725	0.86 (0.83–0.89)	44.7 (37.6–52.1)	60.5 (53.1–67.4)	66.3 (59.0–72.9)	74.2 (67.2–80.1)
4 weeks (29–35 days)	2965	0.85 (0.82–0.87)	35.6 (30.1–41.5)	57.1 (51.2–62.8)	63.3 (57.4–68.8)	74.4 (68.9–79.2)
5 weeks (36–42 days)	2596	0.82 (0.79–0.84)	30.4 (24.1–37.5)	47.9 (40.7–55.2)	52.6 (45.3–59.8)	61.3 (54.0–68.1)
6 weeks (43–49 days)	1276	0.77 (0.72–0.82)	28.1 (19.6–38.3)	39.6 (29.9–50.1)	44.8 (34.7–55.3)	57.3 (46.8–67.2)
**Fetal Medicine Foundation**						
1 week (8–14 days)	156	0.94 (0.89–0.98)	58.3 (36.9–77.2)	75.0 (52.9–89.4)	91.7 (71.6–98.6)	95.8 (76.8–99.8)
2 weeks (15–21 days)	767	0.91 (0.88–0.94)	59.6 (49.2–69.2)	74.7 (64.8–82.7)	84.8 (75.9–91.0)	87.9 (79.4–93.3)
3 weeks (22–28 days)	1725	0.87 (0.84–0.90)	44.7 (37.6–52.1)	61.1 (53.7–68.0)	70.0 (62.9–76.3)	77.9 (71.2–83.4)
4 weeks (29–35 days)	2965	0.88 (0.86–0.90)	45.3 (39.5–51.2)	66.8 (61.0–72.1)	75.1 (69.6–79.9)	79.9 (74.7–84.3)
5 weeks (36–42 days)	2596	0.84 (0.81–0.86)	33.5 (27.0–40.7)	52.6 (45.3–59.8)	64.4 (57.2–71.0)	71.1 (64.1–77.3)
6 weeks (43–49 days)	1276	0.81 (0.77–0.85)	28.1 (19.6–38.3)	42.7 (32.8–53.2)	55.2 (44.7–65.2)	63.5 (53.0–72.9)

## Data Availability

The data analyzed were retrieved from the Miguel Servet University Hospital database.

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
