# Peer review of "Prediction of Late-Onset Small for Gestational Age and Fetal Growth Restriction by Fetal Biometry at 35 Weeks and Impact of Ultrasound–Delivery Interval: Comparison of Six Fetal Growth Standards"

_jcm, 2021, doi:10.3390/jcm10132984_

Round 1
Reviewer 1 Report
Thank you for the opportunity to review this manuscript addressing the predictive ability of estimated percentile weight according to different growth standards by ultrasound at 35 weeks. I also appreciated the fact that the authors tried to determine whether the delivery intervals influence the detection rate of SGA newborns.
A review such as this, helps guide clinicians, patients and their families in detecting any SGA newborns on time, in order to provide the best antenatal care in pregnant women in an effort to decrease any adverse perinatal outcome.
The manuscript is very well-written with little to no need for language editing. The Methodology is very rigorous, and the results are depicted clearly.
The only comment I would like to address is based on the conclusion statement of the authors that the shorter ultrasound delivery interval for te different standards, relates better prediction rates for small gestational age cases. In this manuscript and analyses of the cohort, the authors have not described in each SGA case whether there was any other adverse factor in the ultrasonographic findings, such as any Doppler abnormalities. In cases where there might have been discovered Doppler abnormalities, the SGA fetuses could be considered IUGRs (Intrauterine Growth Restricted fetuses), thus, might be delivered earlier due to their condition, in an effort to avoid a stillbirth. The authors do not mention if there were any inductions of labor for this indication. In my opinion, this could be a confounding factor that should be addressed.
Nonetheless, this is an excellent article that deserves much attention.
Author Response
First, we thank the Reviewer for his/her appreciation of our work and for the suggestions that help us improving the manuscript. We have revised it accordingly and our answers to the Reviewer’s questions are reported below.
Thank you for the opportunity to review this manuscript addressing the predictive ability of estimated percentile weight according to different growth standards by ultrasound at 35 weeks. I also appreciated the fact that the authors tried to determine whether the delivery intervals influence the detection rate of SGA newborns.
A review such as this, helps guide clinicians, patients and their families in detecting any SGA newborns on time, in order to provide the best antenatal care in pregnant women in an effort to decrease any adverse perinatal outcome.
The manuscript is very well-written with little to no need for language editing. The Methodology is very rigorous, and the results are depicted clearly.
The only comment I would like to address is based on the conclusion statement of the authors that the shorter ultrasound delivery interval for te different standards, relates better prediction rates for small gestational age cases. In this manuscript and analyses of the cohort, the authors have not described in each SGA case whether there was any other adverse factor in the ultrasonographic findings, such as any Doppler abnormalities. In cases where there might have been discovered Doppler abnormalities, the SGA fetuses could be considered IUGRs (Intrauterine Growth Restricted fetuses), thus, might be delivered earlier due to their condition, in an effort to avoid a stillbirth.
We thank the reviewer for this comment, we have tried to focus our work on the detection of fetal growth disorders, based on the fetal weight estimated by ultrasound and the weight of the newborn. It is very interesting the comment of IUGRs, which have increased the probability of stillbirth.
As we did not perform Doppler universally (only in cases of estimated fetal weight <10th percentile), we did not study the subgroup of SGAs at delivery with altered Doppler. This is because a significant percentage of SGAs at delivery did not present an estimated fetal weight <10th percentile by ultrasound.
We added a new comment in Material and methods section, 2.2 Estimated percentile weight, line 193.
The authors do not mention if there were any inductions of labor for this indication. In my opinion, this could be a confounding factor that should be addressed.
We agree with this comment, effectively, a small percentage of labors are inductions of labor or cesarean sections programmed by IUGR and they could act as confounding factors in the study. In any case, we add it in weaknesses as the reviewer comments. Similarly, other cases of early termination due to other causes have not been taken into account.
We added a new comment in the discussion section, limitations of the study, line 428.
Nonetheless, this is an excellent article that deserves much attention.
We thank the reviewer for this comment, we hope that this work captures the interest of Journal of Clinical Medicine readers.
Reviewer 2 Report
Thank you for the opportunity to review the paper - Prediction of late-onset small for gestational age and fetal growth restriction by fetal biometry at 35 weeks and impact of ultrasound-delivery interval: comparison of six fetal growth standards. The paper is well-written. The study addresses detection limited to late third trimester detection of SGA to determine APOs.
To the authors, I have the following questions:
- Why was 35 weeks the choice to do ultrasound to detect late onset SGA? Why not 32 weeks?
- Comparing all standards, all AUCs are >0.8. Is 0.87 really better than 0.82 in terms of clinical relevance? How and why? I think it should be discussed in the paper.
- Given the comparison, why do the authors think that non-customized standards offer an advantage over customized standards in predicting APOs? Beyond the numbers, I did not see the rationale, speculative or data-based.
- Once the EPW is determined as SGA at 35 weeks, what is the course of action in terms of fetal surveillance and recommendation for delivery.
Author Response
First, we thank the Reviewer for his/her appreciation of our work and for the suggestions that help us improving the manuscript. We have revised it accordingly and our answers to the Reviewer’s questions are reported below.
Thank you for the opportunity to review the paper - Prediction of late-onset small for gestational age and fetal growth restriction by fetal biometry at 35 weeks and impact of ultrasound-delivery interval: comparison of six fetal growth standards. The paper is well-written. The study addresses detection limited to late third trimester detection of SGA to determine APOs.
To the authors, I have the following questions:
- Why was 35 weeks the choice to do ultrasound to detect late onset SGA? Why not 32 weeks?
In our center we perform the third trimester ultrasound at 35-36 weeks to try to increase the detection of fetal growth alterations, which seems higher than at 32 weeks.
We added a comment to inform this in the Material and methods section, 2.1 study design, line159, and a new reference (McCowan LM, Figueras F, Anderson NH. Evidence-based national guidelines for the management of suspected fetal growth restriction: comparison, consensus, and controversy. Am J Obstet Gynecol 2018;218(2S):S855-S868)
- Comparing all standards, all AUCs are >0.8. Is 0.87 really better than 0.82 in terms of clinical relevance? How and why? I think it should be discussed in the paper.
We agree with the reviewer that although we find statistically significant differences between AUCs in Table 4, all standards shown a good predictive ability to detect SGA at birth. In our conclusions we remarked that “the growth standards showed a similar good predictive ability”. We have softened our conclusions in the abstract, line 40, in this regard.
- Given the comparison, why do the authors think that non-customized standards offer an advantage over customized standards in predicting APOs? Beyond the numbers, I did not see the rationale, speculative or data-based.
We thank the reviewer for this comment. In our analysis, we found that customized models (MSUH and Figueras standards) shown the lowest sensitivities values for different false positive rates (5%-20%). We think that it can be related to the low rate of estimated percentile weight < 10% in our cohort for both standards (8% and 5%, respectively), although Intergrowth 21st showing similar percentage (7.9%) detecting a high percentage of APOs.
In this regard, in the discussion section we have provide this explanation:
“The main reason lies in the greater proportion of 10th percentile EPW for Fetal Medicine Foundation (21.2%) and WHO (12.6%) standards, In any case, with similar proportion of EPW <10, non-customized MSUH and Intergrowth standards show a better APOs predictive ability than customized MSUH and Figueras standards”.
- Once the EPW is determined as SGA at 35 weeks, what is the course of action in terms of fetal surveillance and recommendation for delivery.
Thank you very much for the comment. According to our results, it would be appropriate to raise the ultrasound-estimated weight percentile cutoff point above 10 for fetal growth control. This is because the 10th percentile has been shown to be insufficient and with low predictive capacity for SGAs at delivery and, therefore, fetuses that can potentially even be IUGR before delivery can escape control and thus try to reduce their morbidity and mortality. Our recommendation, in the ultrasound of the third trimester between 35 and 36 weeks, could be to raise the cut-off point at least from 10 to the 20th percentile for strict control of fetal growth.
We added this comment in the section 4.3 Clinical and research implications, line 413.
This manuscript is a resubmission of an earlier submission. The following is a list of the peer review reports and author responses from that submission.